# A Recurrent Inflammatory Myofibroblastic Tumor-like Lesion of the Splenic Capsule in a Kitten: Clinical, Microscopic and Ultrastructural Description

**DOI:** 10.3390/vetsci8110275

**Published:** 2021-11-12

**Authors:** Silvia Ferro, David Chiavegato, Piergiorgio Fiorentin, Valentina Zappulli, Stefano Di Palma

**Affiliations:** 1Department of Biomedicine and Alimentary Science, University of Padova, 35020 Padova, Italy; valentina.zappulli@unipd.it; 2Clinica Veterinaria Arcella, 35133 Padova, Italy; davidchiavegato@yahoo.it (D.C.); piergiorgiofiorentin@yahoo.it (P.F.); 3Preclinical Development, Aptuit (Verona) Srl, an Evotec Company, 37135 Verona, Italy; sdipalma78@gmail.com

**Keywords:** inflammatory myofibroblastic tumor, feline, transmission electron microscopy, spleen

## Abstract

Objective: To describe the findings of an unusual splenic tumor in a kitten. Methods: A grossly multinodular mass arising from the splenic capsule of a 7-month-old male Havana kitten was echographically detected and surgically removed by splenectomy, then analyzed microscopically and ultrastructurally. Results: The mass showed features of an inflammatory myofibroblastic tumor. Treatment and Outcome: Two months after surgical excision, the mass recurred in the same intra-abdominal area but disappeared after 2 months of anti-inflammatory therapy. Follow-up at 18 months after surgery revealed resolution of the disease. Clinical Relevance: Inflammatory myofibroblastic tumor in cats have been rarely reported and are usually in the orbital region. In the present report, an unusual multinodular gross presentation, a recurrence over time, and a favorable clinical course, are described.

## 1. Introduction

Inflammatory myofibroblastic tumor (IMT) is a histologically distinctive lesion in human medicine [1]. This lesion has been reported in the past as atypical pseudosarcoma, atypical myofibroblastic tumor, atypical fibromyxoid tumor, and plasma cell granuloma [2]. The supposed nature of this pathologic entity has changed over time but it is now classified as ‘intermediate’ between a benign and a malignant neoplastic process. The behavior is unpredictable. The disease course is usually benign, but with a strong tendency for recurrence [2,3]. Common sites are lung, abdominal organs, endocrine glands, urogenital tract, orbit, skin, central nervous system, and mesentery [2,4,5,6,7]. In some cases, metastases have been reported [8]. IMT has also been improperly called inflammatory pseudotumor. However, inflammatory pseudotumor is a general/umbrella term including not only IMT but also several other conditions such as reparative lesions; postoperative reactions; pseudosarcomatous myofibroblastic proliferations associated for example with Epstein–barr virus, Herpes virus, HIV, protozoan (Entamoeba) [3], or bacterial infestations *(Mycobacterium, actinomycetes, Nocardia, mycoplasma, Escherichia coli, Klebsiella, Bacillus sphaericus, Pseudomonas, and Helicobacter pylori*); inflammatory-follicular dendritic cell tumors; myofibroblastic neoplasms; and inflammatory fibrosarcoma [1,9]. The clinical presentation is usually characterized by fever [3], anemia, thrombocytosis, polyclonal hyperglobulinemia, reduced weights of abdominal organs, weakness, and abdominal pain [5]. Primary treatment is represented by surgical excision [2] even of the recurrences [5]. When surgery is not possible or incomplete, corticosteroid monotherapy or non-steroidal anti-inflammatory therapy represents an alternative/additional therapy [3]. Histologically, the tumor has a heterogeneous appearance and is composed of a proliferation of myofibroblastic cells, positive for vimentin, cytokeratins, smooth muscle actin, and muscle-specific actin [5], in an inflammatory background composed of plasma cells, lymphocytes, eosinophils, and neutrophils [1].

In veterinary medicine, the disease has been diagnosed as IMT only in a few canine cases, and similar lesions have been described under the name of (inflammatory) pseudotumor in the dog, cat, horse, bushbaby, and brush-tailed porcupine [10,11]. Most common sites of these cases in animals were the orbit and the lung, and most of them were associated with concomitant bacterial infections, such as mycobacteriosis.

In this article, we report the clinical, microscopic, and ultrastructural description of a case of intra-abdominal multinodular recurrent myofibroblastic proliferation with an inflammatory component in a cat.

## 2. Case Presentation

A 7-month-old, cryptorchid male, FIV-FeLV-negative, Havana kitten was presented for mild weakness, inappetence, abdominal distention, and occasional vomit. Familiar history reported the death of a brother-mate for an abdominal mass of unknown origin. Physical examination evidenced a mass at abdominal palpation of the cranial abdomen. An abdominal ultrasonography revealed the presence of multiple, up to 5 cm in diameter, hypoechogenic nodules adjacent to and sharing the blood supply with the spleen. Thoracic radiograph and general blood tests (hematology and biochemistry for liver and kidney) were unremarkable. Cytology of the nodules detected the presence of a slightly atypical spindle cell population in a markedly hematic background and a mixed mainly eosinophilic infiltrate. After 1 week, the cat underwent exploratory laparotomy and subsequent splenectomy. Approximately 15 contiguous, variably-sized (1–5 cm), well-demarcated, soft, red nodules were attached to the splenic capsule on the parietal and visceral surfaces of the distal extremity of the spleen (Figure 1). Grossly, the differential diagnoses included splenic sarcoma, hamartoma, choristoma, or malformation, and, less likely, granulomatous/pyogranulomatous perisplenitis due to Feline Infectious Peritonitis virus (FIPV) or *Mycobaterium* spp.

Soon after surgery, the spleen was fixed with 10% neutral buffered formalin, paraffin embedded, and processed for routine histology and immunohistochemistry. Immunohistochemistry was performed on sections mounted on TOMO® Adhesion Microscope Slides (Matsunami, Osaka, Japan) and dried at 37 °C for 30 minutes. All samples were tested with a semi-automated immunostainer (BenchMark; Ventana Medical Systems, Tucson, AZ, USA), which included dewaxing and rehydration, antigen retrieval with CC1 (Ventana Medical Systems, Tucson, AZ, USA), primary antibody incubation, antigen detection with an ultraView Universal diaminobenzidine (DAB) kit (Ventana Medical Systems, Tucson, AZ, USA) for 8 minutes, and counterstaining with modified Gill hematoxylin for 8 minutes. Finally, slides were manually dehydrated through a graded series of alcohols and mounted (Eukitt mounting medium; Eukitt, Fort Washington, PA, USA). Primary antibody dilutions were performed using a commercial antibody diluent (Ventana Medical Systems, Tucson, AZ, USA). The following antibodies were used: VWF (rabbit Anti-Human Von Willebrand Factor, code A 0082, Dako Glostrup, Denmark), desmin (monoclonal mouse, anti-Human desmin, clone D33, Dako, Glostrup, Denmark), GFAP (monoclonal mouse anti-Human glial fibrillary acid protein, clone 6F2, Histo-Line Lab., Pleasanton, CA, USA), calponin (monoclonal mouse anti-Human calponin, clone CALP, Dako, Glostrup, Denmark), vimentin (monoclonal mouse anti-vimentin, clone V9, Dako, Glostrup, Denmark), panCK (monoclonal mouse anti-Human cytokeratin clones AE1/AE3, Dako, Glostrup, Denmark), SMA (monoclonal mouse anti-Human smooth Muscle Actin, clone 1A4, Dako, Glostrup, Denmark), HLA-dr (monoclonal mouse anti-Human HLA-DR, alpha chain, clone TAL.1B5, Dako, Glostrup, Denmark), and Iba1 (polyclonal rabbit anti Iba1, Wako, Osaka, Japan). Table 1 reports the main details and relative positive controls of the used antibodies. As negative controls, the normal splenic lymphoid tissue (internal negative control) was used for all antibodies, except for vimentin, for which a mammary gland tissue was used. Finally, the paraffin-embedded samples were hydrated and washed in a cacodylate buffer. Subsequently they were postfixed in osmium tetroxide 1% in 0.1M sodium cacodylate buffer for 1 hour at 4 °C, dehydrated in a graded ethanol series, and embedded in an epoxy resin (Sigma-Aldrich, Darmstadt, Germany). Ultrathin sections (60–70 nm) were obtained with an Ultrotome V (LKB) ultramicrotome, counterstained with uranyl acetate and lead citrate and viewed with a Tecnai G2 (FEI) transmission electron microscope (TEM) operating at 100 kV. Images were captured with a Veleta (Olympus Soft Imaging System, Segrate, Italy) digital camera.

Microscopically, the nodules arose exophytically from the splenic capsule with no invasion of the splenic parenchyma. They were mainly composed of a proliferation of spindle cells, arranged in irregular bundles, embedded in a moderate multifocally collagenous or myxoid stroma (Appendix A). Cellular atypia was mild to moderate, and mitotic count was less than 1 per 10 high power field (i.e. 2.37 mm^2^). The spindle cell component was multifocally associated with proliferation of small capillaries, wide cystic spaces filled with red blood cells, and occasional mild hemorrhages. Spindle cells were admixed with disseminated inflammatory cells represented by eosinophils and scattered mononuclear cells resembling histiocytes (Figure 2). The sections stained with Ziehl-Neelsen were negative for the presence of acid-fast bacteria.

Immunohistochemical investigation on the nodular masses revealed diffuse moderate cytoplasmic labeling for vimentin, α-smooth muscle actin (α-SMA) (Figure 3, Appendix A), and desmin of the atypical spindle cell population. The majority of them also showed moderate cytoplasmic immunolabeling for calponin. Immunohistochemistry was negative for GFAP, pan-cytokertain, and Von Willebrand Factor. Furthermore, immunohistochemistry for HLA-dr and ionized calcium binding adapter molecule 1 (Iba1) revealed a positive round cell population admixed with the spindle cell component (Figure 3) and consistent with macrophages/histiocytes.

At TEM, the atypical spindle cells showed cytoplasmic microfilaments referable to contractile filaments (Figure 4, Appendix A). Eosinophils, lymphocytes, plasma cells, and histiocytic cells were also observed. The histological, immunohistochemical, and electron microscopy patterns were compared and found very similar to the features of a canine IMT obtained from the histological archive of the diagnostic service of veterinary pathology of the university of Padova. Based on these findings, a diagnosis of IMT-like lesion was favored.

After 2 months from surgery, the cat was submitted to a follow up visit, he was asymptomatic but a new abdominal nodule near the splenic area was detected via an ultrasound examination. The nodule appeared 13×14 mm in diameter and its architecture did not reveal any pre-existing structure. At the same time, a moderate lymphadenomegaly of the cranial mesenteric, and cranial and caudal colic lymph nodes was detected. Cytological examination of this neoformation confirmed the presence of the same spindle cell population (Figure 5) observed at first cytology of the previously excised masses. The cells were embedded in a hematic background without any lymphoid component. A local recurrence of the lesion was suspected. The owner reclined surgical resection, and the patient underwent medical therapy with 1mg/kg b.i.d. of prednisone for 2 months. At 6 months after surgery, the cat was alive without any clinical signs. Unfortunately, the owner refused any further follow up by diagnostic imaging of the lesion. After 18 months from surgery, the cat died of intestinal venous infarction due to a mesenteric intestinal herniation of probable traumatic origin. Necropsy revealed the absence of any abdominal nodules. An accessory spleen was detected, and its macroscopic and microscopic examination was unremarkable.

## 3. Discussion

The present article reports the unusual presentation and outcome of a myofibroblastic proliferation with an inflammatory component arising from the splenic capsule of a 7-month-old kitten, detected by abdominal ultrasonography, and treated by splenectomy and long-term medical therapy with steroids.

Most splenic masses in cats are represented by primary and metastatic neoplasia, including mastocytoma, lymphoma, myeloproliferative disease, and hemangiosarcoma [12]. Splenic abnormalities are usually investigated by ultrasound-guided fine needle aspirate or fine-needle biopsy, ultrasound-guided core biopsy, surgical core biopsy, or necropsy as characteristic changes are not seen for any of the diseases on imaging alone [13]. In our case, ultrasound-guided fine needle aspirate yielded slightly atypical spindle cells in an eosinophilic inflammatory background. Based on the high reported incidence of splenic neoplasia in cats [12], splenectomy was performed to better characterize the lesion in such a young cat. Clinical presentation, macroscopic appearance, histopathology, immunohistochemistry, and TEM revealed several features shared with human IMT, such as visceral presentation in young patients; multinodular appearance; and proliferation of spindle cells with atypia with a muscular immunophenotype and contractile filaments, admixed with an inflammatory component mainly made of eosinophils. Based on all these findings, a diagnosis of IMT-like lesion was favored. Potential differential diagnosis in our case have been considered, such as inflammatory leiomyosarcoma and inflammatory reactions. However, leiomyosarcomas are usually histologically characterized by cigar-shaped nuclei and no association with an eosinophilic inflammatory component has been reported in the cat, not only in the spleen but also in other locations (including non-visceral). Eosinophilic-rich inflammatory lesions have been reported in the gastrointestinal tract of cats and named “feline gastrointestinal eosinophilic sclerosing fibroplasia”. However, apart from the eosinophilic inflammatory component and immunohistochemical findings (spindle cells were uniformly positive for vimentin and smooth muscle actin), the other histologic features did not match our case (lack of atypia of the spindle cell component), as well as location (gastrointestinal tract and mesenteric lymph nodes),and clinical presentation (all cats were >1 year old) [14]. An exuberant granulation tissue has also been considered in the differential diagnosis for non-neoplastic diseases. However, there were no indications of a previous visceral trauma (e.g. car accident, abdominal surgery) in the reported history of the kitten. For the same reason, an inflammatory myofibroblastic proliferation secondary to an infectious cause was considered unlikely because of the lack of clinical signs, history, and unremarkable blood test supporting the presence of an infectious disease.

Finally, despite local recurrence, medical treatment with long-term steroids after splenectomy appears to be able to control the disease, with an absence of peritoneal nodules on post-mortem examination after 18 months from the initial diagnosis.

## 4. Conclusions

We described the clinical presentation, macroscopical, microscopical, and ultrastructural features of a recurrent, extra-orbital, extra-pulmonary IMT-like lesion in a cat unlikely to be associated with an infectious agent. The distinctive cell population in IMTs, both in humans and in animals, is represented by spindle cells of myofibroblastic origin mixed with an inflammatory population characterized mainly by eosinophils and histiocytes; the latter are typically diffusely disseminated throughout the mass and intermingled with the myofibroblastic cells, while eosinophils are usually also scattered but with variable density throughout the tumor. Compared to other cases described in veterinary medicine, the present case is peculiar for the unusual gross presentation with multiple concurrent nodules and for the recurrence.

## Figures and Tables

**Figure 1 vetsci-08-00275-f001:**
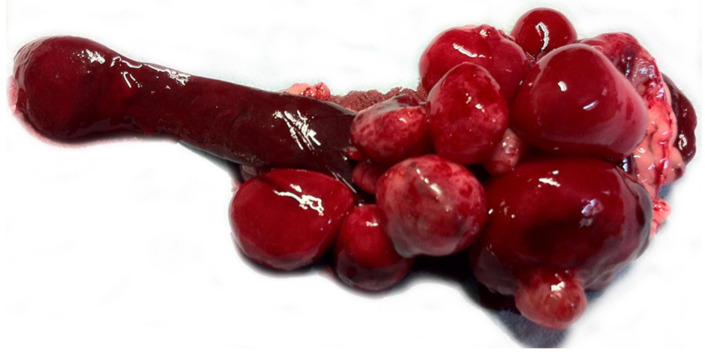
Inflammatory myofibroblastic tumor, spleen, cat. Multiple variably sized reddish nodules arising from the splenic capsule.

**Figure 2 vetsci-08-00275-f002:**
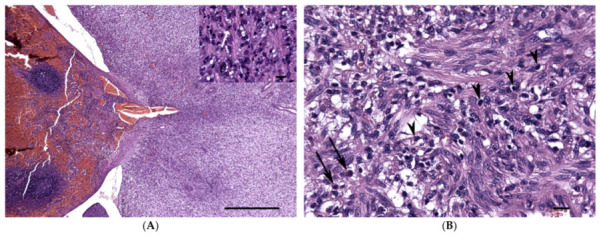
Inflammatory myofibroblastic tumor, spleen, cat. Hematoxylin Eosin. (**A**) On the right side of the image there is a portion of the mass merging from the splenic capsule. Normal splenic parenchyma on the left. Bar = 2.5 mm; magnification 5×. Inset: Higher magnification of the mass composed of mesenchymal spindle cells mixed with few inflammatory cells including some eosinophils (white arrow). Bar = 30 μm, magnification 20×. (**B**) Closer view showing mesenchymal spindle cells embedded in loose fibrillar eosinophilic matrix and organized in small irregular bundles. Disseminated inflammatory cells are also present, as lymphocytes (arrowheads) and eosinophils (black arrows). Bar = 50 μm, magnification 20×.

**Figure 3 vetsci-08-00275-f003:**
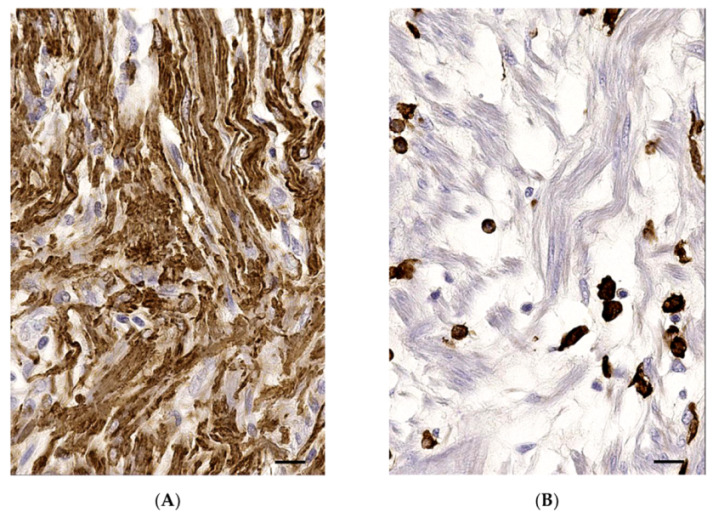
Myofibroblastic inflammatory tumor, spleen, cat. (**A**) Moderate cytoplasmic immunolabeling of the atypical spindle cells. Immunohistochemistry (IHC) for α-SMA. Bar = 50 μm, magnification 20×. (**B**) Marked cytoplasmic immunolabeling of the disseminated round cell population (histiocytes). IHC of Iba1. Bar = 50 μm, magnification 20×.

**Figure 4 vetsci-08-00275-f004:**
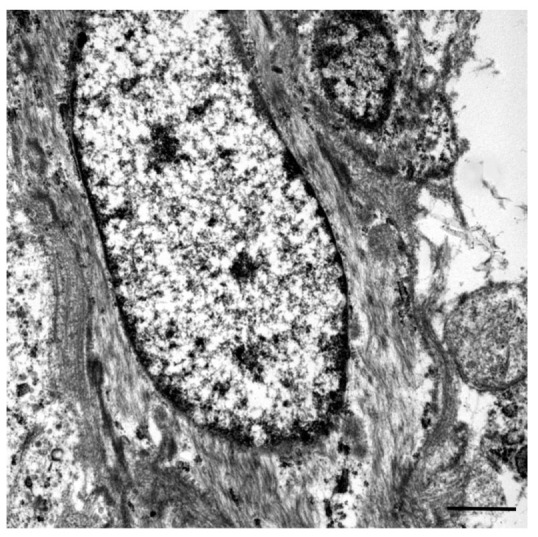
Inflammatory myofibroblastic tumor, abdominal mass, cat. Note the fibrillar cytoplasmic appearance of an atypical spindle cell. Size bar = 1 micron. TEM.

**Figure 5 vetsci-08-00275-f005:**
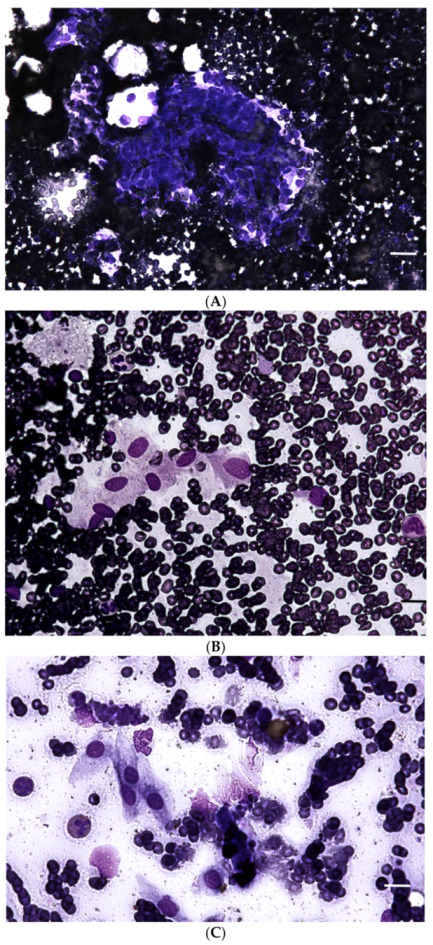
Intra-abdominal inflammatory myofibroblastic tumor, cat. May-Grünwald-Giemsa. Cytological specimen of the recurrence characterized by the presence of spindle to stellate cells visible as aggregates (**A**), size bar = 40 μm, magnification 10×) or single cells (**B**), size bar= 20 μm, magnification 20×). The background is highly hematic. (**C**). First cytology shows the presence of the same population. Size bar= 20 μm, magnification 20×.

**Table 1 vetsci-08-00275-t001:** List of the tested antibodies and synthesis of the used protocol.

Antibody, Brand	Dilution	Antigen Retrieval	Incubation	Amplifier	Antibody, Brand	Positive Control
VWF, Dako	1:1000	30’ at 100 °C	8’	No	VWF, Dako	Internal control (endothelium)
Desmin, Dako	1:50	30’ at 100 °C	16’	No	Desmin, Dako	Muscle
GFAP, Histo-Line Lab	1:50	30’ at 100 °C	24’	No	GFAP, Histo-Line Lab	Brain
Calponin, Dako	1:200	30’ at 100 °C	12’	No	Calponin, Dako	Mammary gland (myoepithelial cells)
Vimentin, Dako	1:150	30’ at 100 °C	28’	No	Vimentin, Dako	Internal control (mesenchymal cells)
PanCK, Dako	1:100	30’ at 100 °C	28’	No	PanCK, Dako	Mammary gland (epithelial cells)
SMA, Dako	1:100	no	12’	No	SMA, Dako	Internal control (vessels walls)
HLA-dr, Dako	1:50	30’ at 100 °C	32’	No	HLA-dr, Dako	Internal control (splenic macrophages)
Iba1, Wako	1:800	8’ at 95 °C	32’	Yes	Iba1, Wako	Internal control (splenic macrophages)

## Data Availability

The data presented in this study are available on request from the corresponding author.

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
