# Peer review of "A Recurrent Inflammatory Myofibroblastic Tumor-like Lesion of the Splenic Capsule in a Kitten: Clinical, Microscopic and Ultrastructural Description"

_vetsci, 2021, doi:10.3390/vetsci8110275_

Round 1

Reviewer 1 Report

In this case report, the authors describe an unusual feline tumor that they interpreted as an inflammatory myofibroblastic tumor (IMT) based on some similarities with this human entity. The paper is rather well written, easy to read and is very well illustrated with beautiful gross, histological, immunohistochemical, cytological and ultrastructural pictures. Of interest are the unusual location (splenic capsule), recurrence of the lesion and the young age of the patient.

General comment: I believe that this nice and unusual case deserves to be published, and I would like to thank the authors for their contribution. However, I have major concerns regarding their final diagnosis and, for this reason, I would recommend revision of the manuscript.

Indeed, IMT is a specific human entity (see Enzinger and Weiss’s Soft Tissue Tumors, 7th ed., p322-330 for a recent reference) and is not a straightforward diagnosis as are squamous cell carcinoma or hemangiosarcoma. Unfortunately, the diagnosis of IMT has been applied questionably to some tumors in animals. Furthermore, there have been confusions (including in this manuscript) between genuine cases of IMT (previously named “inflammatory pseudotumor”) and (inflammatory) pseudotumors as a descriptive diagnosis (for example in the context of mycobacteriosis).

The authors nicely describe a tumor that has an inflammatory component and a muscular/fibroblastic phenotype. While this can lead to a descriptive diagnosis of “inflammatory myofibroblastic tumor”, this is not sufficient to diagnose IMT as an entity, especially based on this single case. This would add further unnecessary confusion to the literature. I also believe that other diagnoses should be considered and discussed, such as an inflammatory leiomyosarcoma/leiomyosarcoma with an inflammatory component, granulation tissue etc.

The authors could also propose an original name for this lesion, as it was done for the orbital pseudotumor of cats mentioned by the authors that have been renamed “Feline Restrictive Orbital Myofibroblastic Sarcoma” (Bell et al., Vet Pathol, 2011, 48(3) 742-750). Alternatively, the lesion could be assigned a more descriptive diagnosis instead of the name of a specific entity: “myofibroblastic neoplasm with inflammatory component”.

As a summary of this general comment, while this case is original, well described and deserves publication, the authors should be cautious with their final diagnosis and other differential diagnoses should be discussed. The discussion is too short and does not address the other differential diagnoses, claiming that other lesions “were not considered” based on human pathology criterions, which is problematic. I would recommend rewriting of the discussion with reference to the most recent papers and textbooks on soft tissue tumors.

Following are more specific comments on the manuscript.

Manuscript: Authors use “spindle” or “fusiform” to describe the cells. Choose only one term.

Title: “XXXX of the splenic capsule in a kitten: clinical, microscopic and ultrastructural description” would be more appropriate, where XXXX corresponds to the chosen diagnosis. I do not think that “recurrent” and, particularly, “multinodular” need to appear in the title.

Line 11: “tumor” instead of “mass”.

Line 19: “favorable” would be more appropriate.

Lines 33-35: Name of agents need to be italicized. Furthermore, I believe that the authors made a confusion here between the etiology of IMT and the etiology of inflammatory pseudotumors in humans (see General comment).

Line 37: “weight loss of abdominal organs” sounds odd. “Reduced weights of abdominal organs” or “atrophy of abdominal organs” sound better.

Line 43: “composed of” instead of “consistent with”

Lines 45-47: References may be more numerous and should correspond to the original papers.

Lines 55-56: “Familial history” instead of “familiar anamnesis”

Lines 59-60: “Thoracic radiographs” instead of “Thorax X-rays”

Line 60: Define “general blood tests”. Does this refer to hematology, biochemistry and/or immunology?

Line 60: “nodules” instead of “neoformations”.

Line 63: Based on the picture, I do not consider “sessile” as the best descriptive term and it is not supported by figure 2 that shows a small pedicle rather than a broad base. Furthermore “sessile” is more traditionally applied to mucosal lesions. I would rather suggest: “Approximatively 15 contiguous variably-sized (1-5 cm), well-demarcated, (soft ?), red nodules were attached to the splenic capsule on the parietal and visceral surfaces of the (distal?) extremity of the spleen.

Lines 65-67: I am not aware of “fibrosarcoma of the splenic capsule” as an entity. I would rather consider “splenic sarcoma” as a broader differential diagnosis. The gross aspect is not supportive of splenic hematoma (would affect the parenchyma and not be restricted to the capsule). Granulomatous/pyogranulomatous perisplenitis due to FIPV or mycobacteriosis can be considered as other differential diagnoses. Considering the young age, I would also consider hamartoma/choristoma/malformation as possible differential diagnoses.

Line 106: would “myxoid stroma” rather than “edematous fibrillar and loose extracellular matrix” be more appropriate?

Line 108: “mitotic count” instead of “mitotic index” (see Meuten et al., Vet Pathol, 2016, Vol. 53(1) 7-9). Mitotic count in 10 HPF would be more precise.

Line 108: Indicate the surface of the HPF (see Meuten et al., Vet Pathol, 2016, Vol. 53(1) 7-9).

Line 111: “eosinophils” instead of “eosinophilic granulocytes”.

Line 113: “acid-fast bacteria” instead of “acid-resistant bacteria”.

Lines 136-139: Such similarities, that cannot be appreciated by the reader, cannot be used as a justification for the final diagnosis.

Line 152: “recurrence” instead of “relapse”.

Line 152: “declined surgical resection” instead of “refused a new surgery”.

Line 154: “signs” instead of “symptoms”.

Line 156: With a mesenteric intestinal herniation, wouldn’t “intestinal venous infarction” be more appropriate instead of “severe hemorrhagic-necrotic enteritis”?

Line 166: “unusual” instead of “peculiar”

Lines 180-182: See General comment.

Line 196: “so far” instead of “until this moment”.

Lines 198-201: “myofibroblastic” instead of “miofibroblastic”, several occurrences.

Kind regards,

Author Response

In this case report, the authors describe an unusual feline tumor that they interpreted as an inflammatory myofibroblastic tumor (IMT) based on some similarities with this human entity. The paper is rather well written, easy to read and is very well illustrated with beautiful gross, histological, immunohistochemical, cytological and ultrastructural pictures. Of interest are the unusual location (splenic capsule), recurrence of the lesion and the young age of the patient.

General comment: I believe that this nice and unusual case deserves to be published, and I would like to thank the authors for their contribution. However, I have major concerns regarding their final diagnosis and, for this reason, I would recommend revision of the manuscript.

Indeed, IMT is a specific human entity (see Enzinger and Weiss’s Soft Tissue Tumors, 7th ed., p322-330 for a recent reference) and is not a straightforward diagnosis as are squamous cell carcinoma or hemangiosarcoma. Unfortunately, the diagnosis of IMT has been applied questionably to some tumors in animals. Furthermore, there have been confusions (including in this manuscript) between genuine cases of IMT (previously named “inflammatory pseudotumor”) and (inflammatory) pseudotumors as a descriptive diagnosis (for example in the context of mycobacteriosis).

The authors nicely describe a tumor that has an inflammatory component and a muscular/fibroblastic phenotype. While this can lead to a descriptive diagnosis of “inflammatory myofibroblastic tumor”, this is not sufficient to diagnose IMT as an entity, especially based on this single case. This would add further unnecessary confusion to the literature. I also believe that other diagnoses should be considered and discussed, such as an inflammatory leiomyosarcoma/leiomyosarcoma with an inflammatory component, granulation tissue etc.

The authors could also propose an original name for this lesion, as it was done for the orbital pseudotumor of cats mentioned by the authors that have been renamed “Feline Restrictive Orbital Myofibroblastic Sarcoma” (Bell et al., Vet Pathol, 2011, 48(3) 742-750). Alternatively, the lesion could be assigned a more descriptive diagnosis instead of the name of a specific entity: “myofibroblastic neoplasm with inflammatory component”.

As a summary of this general comment, while this case is original, well described and deserves publication, the authors should be cautious with their final diagnosis and other differential diagnoses should be discussed. The discussion is too short and does not address the other differential diagnoses, claiming that other lesions “were not considered” based on human pathology criterions, which is problematic. I would recommend rewriting of the discussion with reference to the most recent papers and textbooks on soft tissue tumors.

Following are more specific comments on the manuscript.

A: we would like to thank the reviewer for the valuable considerations about how to name this entity. In order to avoid too much changes to the structure of the paper, which went under a minor revision for the decision of the editor, we finally decided to refer to the lesion, in this paper, as an IMT-like, based on the similarity of the microscopic and ultrastructural aspects observed in humans and other animals. We still consider that this could be the most likely differential diagnosis, in any case the entity with the most similarities.

Manuscript: Authors use “spindle” or “fusiform” to describe the cells. Choose only one term.

A: we chose and used only the term spindle.

Title: “XXXX of the splenic capsule in a kitten: clinical, microscopic and ultrastructural description” would be more appropriate, where XXXX corresponds to the chosen diagnosis. I do not think that “recurrent” and, particularly, “multinodular” need to appear in the title.

A: We changed the title as follow: “A recurrent inflammatory myofibroblastic tumor-like lesion of the splenic capsule in a kitten: clinical, microscopic and ultrastructural description”. We would like to keep the word recurrent to highlight the peculiarity of this case. We agree that the word multinodular is instead not necessary and we removed it.

Line 11: “tumor” instead of “mass”.

A: adjusted as suggested.

Line 19: “favorable” would be more appropriate.

A: adjusted as suggested, thank you.

Lines 33-35: Name of agents need to be italicized. Furthermore, I believe that the authors made a confusion here between the etiology of IMT and the etiology of inflammatory pseudotumors in humans (see General comment).

A: we changed to Italic, thank you for notice the mistake.

We also tried to clarify the IMT and pseudotumor etiologies at lines 32-39

Line 37: “weight loss of abdominal organs” sounds odd. “Reduced weights of abdominal organs” or “atrophy of abdominal organs” sound better.

A: adjusted as suggested.

Line 43: “composed of” instead of “consistent with”

A: adjusted as suggested, thank you.

Lines 45-47: References may be more numerous and should correspond to the original papers.

A: a few recent citations has been added.

Lines 55-56: “Familial history” instead of “familiar anamnesis”

A: adjusted as suggested.

Lines 59-60: “Thoracic radiographs” instead of “Thorax X-rays”

A: adjusted as suggested

Line 60: Define “general blood tests”. Does this refer to hematology, biochemistry and/or immunology?

A: we specified in the text the performed test type.

Line 60: “nodules” instead of “neoformations”.

A: adjusted as suggested.

Line 63: Based on the picture, I do not consider “sessile” as the best descriptive term and it is not supported by figure 2 that shows a small pedicle rather than a broad base. Furthermore “sessile” is more traditionally applied to mucosal lesions. I would rather suggest: “Approximatively 15 contiguous variably-sized (1-5 cm), well-demarcated, (soft ?), red nodules were attached to the splenic capsule on the parietal and visceral surfaces of the (distal?) extremity of the spleen.

A: we confirm and added the consistency and the location of the nodules on the spleen.

Lines 65-67: I am not aware of “fibrosarcoma of the splenic capsule” as an entity. I would rather consider “splenic sarcoma” as a broader differential diagnosis. The gross aspect is not supportive of splenic hematoma (would affect the parenchyma and not be restricted to the capsule). Granulomatous/pyogranulomatous perisplenitis due to FIPV or mycobacteriosis can be considered as other differential diagnoses. Considering the young age, I would also consider hamartoma/choristoma/malformation as possible differential diagnoses.

A: we agree and thank the reviewer for the suggestions. Actually, due to the peculiar aspect of the lesion, a list of realistic differential diagnosis based only on the macroscopic aspect was difficult to enunciate.

Line 106: would “myxoid stroma” rather than “edematous fibrillar and loose extracellular matrix” be more appropriate?

A: we chose to refer to a collagenous or myxoid stroma, because it varied inside the tumor.

Line 108: “mitotic count” instead of “mitotic index” (see Meuten et al., Vet Pathol, 2016, Vol. 53(1) 7-9). Mitotic count in 10 HPF would be more precise.

A: you are right, sorry for the mistake. We changed to mitotic count.

Line 108: Indicate the surface of the HPF (see Meuten et al., Vet Pathol, 2016, Vol. 53(1) 7-9).

A: the surface for the mitotic count has been added as required.

Line 111: “eosinophils” instead of “eosinophilic granulocytes”.

A: adjusted as suggested.

Line 113: “acid-fast bacteria” instead of “acid-resistant bacteria”.

A: adjusted as suggested.

Lines 136-139: Such similarities, that cannot be appreciated by the reader, cannot be used as a justification for the final diagnosis.

A: the sentence has been changed as follow: “Based on these findings, a diagnosis of IMT-like”

Line 152: “recurrence” instead of “relapse”.

A: adjusted as suggested, thank you.

Line 152: “declined surgical resection” instead of “refused a new surgery”.

A: adjusted as suggested.

Line 154: “signs” instead of “symptoms”.

A: adjusted as suggested.

Line 156: With a mesenteric intestinal herniation, wouldn’t “intestinal venous infarction” be more appropriate instead of “severe hemorrhagic-necrotic enteritis”?

A: adjusted as suggested.

Line 166: “unusual” instead of “peculiar”

A: adjusted as suggested.

Lines 180-182: See General comment.

A: the discussion has been changed after renaming the lesion.

Line 196: “so far” instead of “until this moment”.

A: this line has been removed, since it has no more sense after changing the name of the lesion.

Lines 198-201: “myofibroblastic” instead of “miofibroblastic”, several occurrences.

A: adjusted, thank you.

Reviewer 2 Report

The present manuscript deals with a multinodular inflammatory myofibroblastic tumor at the splenic capsule in a cat. It is an interesting, straightforward description, correctly performed. It contains interesting data, useful for treating similar cases.

Minor remarks:

Line 12: has been to be changed by was

Lines 83-96: I always prefer a table for exposing the complete set of data on antibodies for immunohistochemistry, I think it is much more clear. I would suggest the authors to introduce such a table, but this is not vital for the manuscript: the information needed is already there. I leave the decision to the authors.

Line 173: Remove the word pathognomonic and change it for significant or something similar. Nothing on earth is pathognomonic: there is always an alternative. I teach my students to eliminate this word from their brains.

Line 188: How do you know the tumor was “non-associated to an infectious agent”. It really looks like but a sentence such as “much likely” would be acknowledged here. Authors never performed a bacteriologic study to rule out this possibility. They only exclude ZN+ bacteria but this is not enough.

Line 196. The sentence “At our knowledge no recurrences have been described until this moment in animals.” Would be better finished with “with this tumor” or something similar.

Author Response

The present manuscript deals with a multinodular inflammatory myofibroblastic tumor at the splenic capsule in a cat. It is an interesting, straightforward description, correctly performed. It contains interesting data, useful for treating similar cases.

Minor remarks:

Line 12: has been to be changed by was

A: adjusted as suggested.

Lines 83-96: I always prefer a table for exposing the complete set of data on antibodies for immunohistochemistry, I think it is much more clear. I would suggest the authors to introduce such a table, but this is not vital for the manuscript: the information needed is already there. I leave the decision to the authors.

A: a table has been added as suggested, thank you.

Line 173: Remove the word pathognomonic and change it for significant or something similar. Nothing on earth is pathognomonic: there is always an alternative. I teach my students to eliminate this word from their brains.

A: we would like to thank the reviewer for sharing his/her opinion. Despite having taught us to say pathognomonic, we agree with the reviewer and we happily changed to ‘characteristic’.

Line 188: How do you know the tumor was “non-associated to an infectious agent”. It really looks like but a sentence such as “much likely” would be acknowledged here. Authors never performed a bacteriologic study to rule out this possibility. They only exclude ZN+ bacteria but this is not enough.

A: the reviewer is right, the sentence has been adjusted as suggested, thank you.

Line 196. The sentence “At our knowledge no recurrences have been described until this moment in animals.” Would be better finished with “with this tumor” or something similar.

A: since a change of the diagnosis has been required from the reviewer 1, this sentence has no more any sense and the line has been then removed.

This manuscript is a resubmission of an earlier submission. The following is a list of the peer review reports and author responses from that submission.